# Effects of Sex on the Susceptibility for Atrial Fibrillation in Pigs with Ischemic Heart Failure

**DOI:** 10.3390/cells12070973

**Published:** 2023-03-23

**Authors:** Valerie Pauly, Julia Vlcek, Zhihao Zhang, Nora Hesse, Ruibing Xia, Julia Bauer, Simone Loy, Sarah Schneider, Simone Renner, Eckhard Wolf, Stefan Kääb, Dominik Schüttler, Philipp Tomsits, Sebastian Clauss

**Affiliations:** 1Grosshadern Campus, Department of Medicine I, University Hospital Munich, Ludwig-Maximilians-University (LMU), Marchioninistrasse 15, D-81377 Munich, Germany; 2German Center for Cardiovascular Research (DZHK), Partner Site Munich, Munich Heart Alliance, D-81377 Munich, Germany; 3Institute of Surgical Research at the Walter-Brendel-Centre of Experimental Medicine, University Hospital Munich, LMU Munich, Marchioninistrasse 68, D-81377 Munich, Germany; 4Interfaculty Center for Endocrine and Cardiovascular Disease Network Modelling and Clinical Transfer (ICONLMU), LMU Munich, Feodor-Lynen-Strasse 19, D-81377 Munich, Germany; 5Chair for Molecular Animal Breeding and Biotechnology, Gene Center and Department of Veterinary Sciences, LMU Munich, Feodor-Lynen-Strasse 25, D-81377 Munich, Germany; 6Center for Innovative Medical Models (CiMM), Department of Veterinary Sciences, LMU Munich, Hackerstrasse 27, D-85764 Oberschleissheim, Germany; 7German Center for Diabetes Research (DZD), Ingolstädter Landstrasse 1, D-85764 Neuherberg, Germany; 8Laboratory for Functional Genome Analysis (LAFUGA), Gene Center, Grosshadern Campus, LMU Munich, Feodor-Lynen-Stasse 25, D-81377 Munich, Germany

**Keywords:** atrial fibrillation, atrial arrhythmias, electrical remodeling, structural remodeling, sex differences, large animal model, pig model, translational medicine

## Abstract

Atrial fibrillation (AF) is the most prevalent arrhythmia, often caused by myocardial ischemia/infarction (MI). Men have a 1.5× higher prevalence of AF, whereas women show a higher risk for new onset AF after MI. However, the underlying mechanisms of how sex affects AF pathophysiology are largely unknown. In 72 pigs with/without ischemic heart failure (IHF) we investigated the impact of sex on ischemia-induced proarrhythmic atrial remodeling and the susceptibility for AF. Electrocardiogram (ECG) and electrophysiological studies were conducted to assess electrical remodeling; histological analyses were performed to assess atrial fibrosis in male and female pigs. IHF pigs of both sexes showed a significantly increased vulnerability for AF, but in male pigs more and longer episodes were observed. Unchanged conduction properties but enhanced left atrial fibrosis indicated structural rather than electrical remodeling underlying AF susceptibility. Sex differences were only observed in controls with female pigs showing an increased intrinsic heart rate, a prolonged QRS interval and a prolonged sinus node recovery time. In sum, susceptibility for AF is significantly increased both in male and female pigs with ischemic heart failure. Differences between males and females are moderate, including more and longer AF episodes in male pigs and sinus node dysfunction in female pigs.

## 1. Introduction

Atrial fibrillation (AF) is the most prevalent cardiac arrhythmia worldwide: over 37.5 million people were diagnosed with AF and nearly 287,000 deaths were associated with AF in 2017, with predictions estimating that both prevalence and incidence will continue to increase drastically in the next years [1].

Myocardial ischemia is an independent risk factor for AF and both diseases often co-exist [2]. Although the overall prevalence of AF is estimated to be 1.5 times higher in men, women seem to be at higher risk of suffering from new onset AF after myocardial infarction [2]. However, despite these known differences, the impact of sex on AF is still incompletely understood, especially regarding the clinical presentation and evaluation of AF in women [3]. Multiple studies report that women have been underrepresented in clinical trials investigating AF in the past [4,5]. In particular, an analysis of all studies cited by the “2020 Canadian Cardiovascular Society/Canadian Heart Rhythm Society Comprehensive Guidelines for the Management of Atrial Fibrillation” showed that only 39.1% of the study population were women [3,6]. In recent years, research began setting a greater focus on investigating differences between men and women concerning pathophysiology, treatment, and prevention [7]. It has been observed that treatment strategies and outcome of cardiovascular diseases are influenced by the gender of patients: women less often receive electrical cardioversion or catheter ablation upon the diagnosis of AF [8] and a nationwide study in Germany showed that the in-hospital mortality of patients with AF, who underwent atrial ablation or atrial appendage occlusion therapy, is also associated with the female sex [9].

Given that, it is clearly necessary to consider the impact of sex on electrophysiology and arrhythmogenesis in basic, translational, and clinical research.

Rodent models have been used to investigate sex-based differences in cardiovascular disease development [10], but there is still a lack of inclusion of sex as a biological variable in animal models in cardiovascular research as most studies use exclusively male or female animals or did not specify the sex at all [11,12]. Although the necessity of reporting sex differences in clinical trials has increased, these requirements have not been converted to preclinical research [12]. Therefore, we wanted to investigate possible biological sex-related differences in heart function, electrophysiology and arrhythmogenicity in a translational porcine model of atrial fibrillation (AF) in the context of ischemic heart failure (IHF), which we have previously characterized [13].

## 2. Materials and Methods

### 2.1. Animals

All animal experiments were conducted in accordance with the “Guide for the Care and Use of Laboratory Animals” and were approved by the *Regierung von Oberbayern* (ROB-55.2-2631.Vet_02-10-130, ROB-55.2-2532.Vet_02-15-209 and ROB-55.2-2532.Vet_02-18-69) at the Institute of Surgical Research at the Walter-Brendel-Centre of Experimental Medicine, Munich, Germany. The animals included in this work were bred in research facilities of the *Landwirtschaftliche Forschungsstation Thalhausen*, Technical University of Munich (TUM), Kranzberg, Germany, the *Moorversuchsgut* and the Center for Innovative Medical Models (CiMM), Ludwig-Maximilians-University (LMU), Oberschleissheim, Germany, and *Lehr-und Versuchsgut der LMU*, Oberschleissheim, Germany.

In total, 72 domestic swine aged 3–6 months with an average weight of 66.1 ± 2.9 kg were included in this analysis. We studied four groups: (1) Male control pigs (63.3 ± 5.1 kg, *n* = 22), (2) Female control pigs (69.7 ± 5.2 kg, *n* = 22), (3) Male IHF pigs (65.6 ± 6.5 kg, *n* = 15) and (4) Female IHF pigs (65.1 ± 7.9 kg, *n* = 13). Inclusion in either the CTL or the IHF group was random. Ischemic heart failure was induced by occlusion of the left anterior descending artery (LAD) for 90 min as previously described [13,14]. After 30 days, IHF pigs underwent the same experimental procedures as age- and weight-matched controls. The pigs were sacrificed after in vivo measurements and hearts were removed for further histological experiments.

### 2.2. Anesthesia and Surgical Preparation

Experimental procedures were recently described in detail [13,14]. In short, all pigs were given seven days of acclimatization in the facility to avoid any stress-induced alterations of experimental results. On the day of surgery, the pigs were sedated by intramuscular (IM) injection of ketamine (20 mg/kg, Zoetis, Berlin, Germany) and azaperone (10 mg/kg, Elanco, Bad Homburg, Germany) in the lateral cervical musculature behind the ear in combination with atropine (0.05 mg/kg, Braun, Melsungen, Germany) to reduce salivation during the endotracheal intubation. Once sedated, an intravenous (IV) access was placed in the outer auricular vein followed by IV injection of midazolam (0.5 mg/kg, Braun, Melsungen, Germany), propofol (0.5 mg/kg/min, Fresenius, Bad Homburg, Germany), and fentanyl (0.05 mg/kg, Dechra, Aulendorf, Germany) to maintain general anesthesia and analgesia at a level of surgical tolerance throughout the experimental procedure.

After intubation and preparation of the surgical site in dorsal recumbency, 8F and 9F sheaths (Cordis, Norderstedt, Germany) were surgically introduced into the right external jugular vein and the right carotid artery. The LAD was occluded by an angioplasty balloon (Merit Medical, South Jordan, UT, USA) placed distal of the first diagonal branch. To prevent ventricular tachyarrhythmias, 150 mg/kg amiodarone (Hikma Pharma, Martinsried, Germany) were administered upon LAD occlusion. The location of the inflated balloon was intermittently confirmed by X-Ray (Ziehm Imaging, Nuremberg, Germany). Furthermore, all vital parameters were closely monitored during 90 min of occlusion and following reperfusion. If the pigs remained stable after the reperfusion period, all catheters and sheaths were removed, and blood vessels were ligated. The surgical site was then thoroughly closed and bandaged, anesthesia was stopped, and pigs were transferred to the housing facility where they were monitored for 30 days.

### 2.3. Assessment of the Ejection Fraction

Using a 6F or 7F Pigtail catheter (Cordis, Norderstedt, Germany), laevocardiography of the left ventricle (LV) was performed to assess the ejection fraction in the left anterior-oblique (LAO) angulation at 30° and anterior-posterior (AP) angulation at 0° at a paced heart rate of 130 bpm.

### 2.4. 12-Lead Electrocardiogram and Electrophysiological Study

12-lead surface electrocardiograms (ECGs) were recorded to measure heart rate (bpm), P wave duration (ms), PR interval (ms), QRS interval (ms) and QT interval (ms). For electrophysiological (EP) studies, a multipolar electrophysiology catheter (Abbott, Eschborn, Germany) was placed at the high right atrium to allow atrial stimulation (at 2× pacing threshold) to assess sinus node recovery time (SNRT), atrioventricular conduction properties, as well as atrial and atrioventricular refractory periods.

The sinus node recovery time (SNRT) is defined as the interval between the last atrial stimulus and the first intrinsic P wave following 30 s of atrial stimulation. SNRT was evaluated at paced cycle lengths of 500 ms, 450 ms, and 400 ms, and was corrected for the intrinsic basic cycle length (SNRT/BCL). The SNRT/BCL is calculated by multiplying the quotient of the SNRT and the BCL by 100. Atrioventricular conduction properties were assessed by measuring the Wenckebach cycle length (WB), the 2:1 conduction cycle length (2:1 CL), the atrioventricular effective refractory period (AVERP) and the atrial effective refractory period (AERP). We determined the Wenckebach point by progressively shortening the atrial pacing cycle length from 500 ms in decrements of 10 ms until there was no longer a 1:1 AV conduction of the atrial signal. The 2:1 conduction cycle length was assessed in the same manner. The effective refractory periods of the atrium and the AV node were assessed by series of seven fixed stimulations at each basic cycle length (500 ms, 450 ms, 400 ms, 350 ms, 300 ms, and 250 ms, respectively) followed by a decrementally coupled, premature, eighth stimulus. AERP and AVERP were defined as the cycle lengths of the premature stimulus at which the atrial signal did not induce a P wave (AERP) or was not conducted to the ventricles (AVERP).

Finally, arrhythmias were induced by burst pacing in the high right atrium for 6 s. Atrial Fibrillation (AF) was defined as an atrial arrhythmia with irregular RR intervals lasting at least 10 s.

### 2.5. Assessment of Structural Remodeling

Structural remodeling was assessed by quantifying interstitial fibrosis in the left atrium and left ventricle (remote of infarcted area). Paraffin-embedded tissue samples were cut at a thickness of 5 µm and stained according to the Masson’s trichrome protocol (Masson-Goldner-Trichrome Staining Kit 3459.1, Carl Roth GmbH + Co. KG, Karlsruhe, Germany). Ten random, non-overlapping images per region were taken using a high-resolution microscope (DM6 B, Leica Mikrosysteme Vertrieb GmbH, Wetzlar, Germany) at a 40× magnification. Interstitial fibrosis was quantified by a blinded observer.

### 2.6. Statistical Analysis

All statistical analyses were performed using GraphPad Prism 8.0.1. Data are presented as Mean ± SEM. The standard error of the mean (SEM) was calculated dividing the standard deviation by the square root of the sample size (*n*) of each data set. Differences between two cohorts were calculated by applying the non-parametric Mann-Whitney-U Test. Fisher’s-Exact Test was used to compare categorical values (presented as percentages). Figures were designed using GraphPad Prism 8.0.1 Individual values of each pig are shown as grey circles. Results were considered statistically significant with *p*-values less than 0.05.

## 3. Results

### 3.1. Induction of Ischemic Heart Failure

In IHF a significant reduction of the ventricular ejection fraction (EF) was observed (Figure 1A and Figure 2A). In male IHF pigs the EF was significantly lower than in male control pigs (EF 0° AP CTL vs. IHF: 59.6 ± 2.5% vs. 22.9 ± 4.1%, *** *p* < 0.001; Figure 1A; 30° LAO: 59.0 ± 2.2% vs. 35.4 ± 3.2%, *** *p* < 0.001; Figure 2A). Consistently, in female pigs the EF was significantly reduced in IHF animals as well (EF 0° AP CTL vs. IHF: 57.2 ± 2.8% vs. 26.8 ± 4.2%, *** *p* < 0.001; Figure 1A; 30° LAO: 58.6 ± 3.0% vs. 31.8 ± 3.9%, *** *p* < 0.001; Figure 2A). No differences were observed between male and female pigs. Hemodynamic measurements obtained from right heart catheterization using a Swan-Ganz catheter (Edwards Lifesciences, Irvine, CA, USA) indicated an increase in pulmonary arterial pressure (PAP), pulmonary capillary wedge pressure (PCWP) and right arterial pressure (RAP) in both IHF groups compared to their respective control groups (Appendix A).

### 3.2. ECG

ECG analysis revealed two significant differences between male and female control pigs, which disappeared in IHF: female control pigs showed a significantly lower heart rate compared to male control pigs (male vs. female: 95.1 ± 6.1 bpm vs. 77.7 ± 5.4 bpm, * *p* < 0.05; Figure 3A) and a significantly prolonged QRS interval (male vs. female: 68.4 ± 2.1 ms vs. 75.3 ± 2.8 ms, * *p* < 0.05; Figure 3D). Other ECG parameters such as P wave duration (Figure 3B), PR interval (Figure 3C), and QT_C_ interval (Figure 3E) did not differ between male and female control pigs. In pigs with ischemic heart failure, no differences were observed in these ECG parameters between males and females or compared to their respective controls (Figure 3).

### 3.3. Arrhythmia Inducibility

The overall inducibility of AF was assessed as the percentage of pigs with AF episodes longer than 10 s among all pigs of each group (Figure 4A). IHF significantly increased the susceptibility for AF in both sexes with 42.9% of the male pigs and 46.2% of the female pigs showing AF (** *p* < 0.01, compared to sex-matched control pigs, respectively; Figure 4A). Furthermore, in both sexes with IHF, a significantly higher number of burst stimulations resulted in AF compared to control pigs (CTL vs. IHF: 2.2% vs. 12.1% in males, *** *p* < 0.001; and 0.4% vs. 4.9% in females, ** *p* < 0.01; Figure 4B). As a result, the average AF burden was significantly increased in IHF pigs compared to control pigs in both sexes (CTL vs. IHF: 6.2 ± 6.2 s vs. 40.6 ± 22.9 s in males, * *p* < 0.05; and 0.9 ± 0.9 s vs. 8.9 ± 3.3 s in females, ** *p* < 0.01; Figure 4C). A similar number of male and female pigs both in control and IHF groups developed AF; however, in male pigs a higher number of stimulations resulted in AF with a statistically significant difference in IHF animals (male vs. female: 12.1% vs. 4.9% in IHF pigs, * *p* < 0.05; and 2.2% vs. 0.4% in control pigs, *p* = 0.1116; Figure 4B). We observed a trend towards a lower AF burden in female pigs, however, these differences were not statistically significant (male vs. female: 40.6 ± 22.9 s vs. 8.9 ± 3.3 s in IHF pigs, *p* = 0.5453; and 6.2 ± 6.2 s vs. 0.9 ± 0.9 s in control pigs, *p* = 0.7445; Figure 4C).

### 3.4. Electrophysiological Studies (EPS)

The uncorrected sinus node recovery times measured at pacing cycle lengths of 500 ms, 450 ms, and 400 ms showed significant differences between male and female control pigs (Appendix A). Female control pigs showed a significantly longer sinus node recovery time to basic cycle length quotient (SNRT/BCL) at all stimulated cycle lengths compared to male control animals (male vs. female: at 500 ms: 119.6 ± 4.3% vs. 152.6 ± 9.7%, * *p* < 0.05; at 450 ms: 120.5 ± 4.3% vs. 159.1 ± 10.7%, ** *p* < 0.01; at 400 ms: 119.9 ± 4.5% vs. 157.9 ± 10.0%, *** *p* < 0.001; Figure 5A–C). A similar trend between the two sexes could be observed in IHF pigs, but only at a stimulated cycle length of 400 ms the difference was statistically significant (male vs. female: 123.6 ± 8.8% vs. 172.5 ± 22.1%, * *p* < 0.05; Figure 5C). Between control and IHF animals of each respective sex, however, no difference was observed.

To determine atrioventricular (AV) conduction properties we measured the Wenckebach cycle length and the 2:1 AV conduction cycle length, but could not observe any statistically significant differences, either between males and females or between control and IHF pigs (Figure 6).

The effective refractory period of the AV node (AVERP) was significantly prolonged in female controls compared to male controls at a basic cycle length of 450 ms (male vs. female: 241.8 ± 9.1 ms vs. 268.7 ± 10.5 ms, * *p* < 0.05; Figure 7B), whereas the other stimulated basic cycle lengths did not show any differences (Figure 7A,C–F). In pigs with IHF, no difference was observed between males and females nor compared to their respective controls (Figure 7). Similarly, we did not see any differences between sexes or between controls and IHF pigs regarding AERP (Figure 8).

### 3.5. Assessment of Structural Remodeling

Histologic analysis revealed a significant increase in left atrial (LA) interstitial fibrosis in both female and male IHF pigs compared to controls (CTL vs. IHF: 5.0 ± 1.1% vs. 11.3 ± 1.8% in males, * *p* < 0.05; and 3.7 ± 0.7% vs. 11.0 ± 2.2% in females, ** *p* < 0.01; Figure 9A). However, there were no significant differences between males and females of CTL or IHF animals. In the left ventricle (LV), IHF resulted in significantly increased fibrosis only in females (CTL vs. IHF: 2.6 ± 0.7% vs. 4.5 ± 0.4%, * *p* < 0.05; Figure 10A), but not in males (CTL vs. IHF: 4.0 ± 0.7% vs. 5.0 ± 0.8%, *p* = 0.3864; Figure 10A). Within each sex, there were no statistical differences between the control and IHF group (male vs. female: *p* = 0.1423 in CTL; *p* = 0.7544 in IHF).

## 4. Discussion

In the presented study, we investigated the potential impact of sex on susceptibility for atrial fibrillation (AF) using a pig model of ischemic heart failure (IHF), and observed a significantly increased vulnerability for AF in IHF pigs of both sexes.

Although sex-related differences in patients with AF are well known, it has not been fully elucidated *how* sex affects AF pathophysiology [8,15]. There is a strong body of evidence suggesting that pre-menopausal women have a lower risk of cardiovascular diseases compared to men, indicating a strong effect of sex hormones on cardiac pathophysiology [16]. This is further supported by data showing that in post-menopausal women the concentration of estrogens, especially estradiol, decrease to the same basal level as in men [17] which is paralleled by an increase in cardiovascular risk compared to age-matched men [18]. Nevertheless, it has been shown that even prepubertal boys and girls have different blood concentrations of estrone (E1) and estradiol (E2) [19], indicating that sex hormones may already play a role in the development of children and contribute to a biological dimorphism even in the young. Mechanistically, sex hormones act via specific receptors for estrogen (ERα and ERβ) and androgen isoforms [16,20], which are phosphorylated in a sex-specific manner in cardiomyocytes of men and women [21]. Myocardial calcium handling and, thus, cardiac electrophysiology are also directly affected by estrogens and androgens [21,22,23]. Furthermore, some studies have shown modulatory effects of sex-steroids on mRNA levels of ion channels [20,23]. One study has recently shown that there is a significant difference in the resting membrane potential (RMP) between men and women, possibly due to fewer inward rectifier channels in women, thus resulting in a slower conduction in the atria of women [24]. This could also be a potential mechanistic explanation for the lower incidence of AF in women [24].

Numerous animal models have been used to evaluate sex differences in cardiac electrophysiology and arrhythmogenesis [8,15,20,22,25]. However, most studies so far have mainly focused on the ventricles, but not on the atria [8,15,20,22,26]. In mice—although they are commonly used in arrhythmia research in general [27]—only a few studies evaluated sex effects on atrial electrophysiology or arrhythmogenesis and revealed inconsistent results. In regard to ECG parameters, some studies have shown longer QRS durations in female mice compared to male mice [28,29]. In other studies, however, no difference was observed [30]. One study also demonstrated a prolonged PR interval in female mice [30], whereas others did not report such a difference [28,29]. Male mice are more susceptible to develop AF induced by programmed electrical stimulation, an effect that is abolished by orchiectomy, which further emphasizes the role of sex hormones [31,32]. Potential mechanisms described include altered calcium homeostasis (increased calcium transient amplitude, more frequent spontaneous calcium releases, faster decay time, higher Na^+^Ca^2+^ exchanger current density and a lower L-type calcium current density in male mice) resulting in enhanced triggered activity in male mice [31] and connexin lateralization [32]. Regarding atrial action potential shape/duration or potassium currents, however, no differences were observed [32]. In contrast, other studies demonstrated no differences between male and female mice in regard to intracardiac conduction properties, or susceptibility to arrhythmias [28,29]. These inconsistencies, which also include contradictory reports on ion channel distribution, are most likely due to different mouse strains or ages studied, as these aspects have been demonstrated as key determinants in murine electrophysiology [25,33,34,35].

Similarly, in other species sex differences have been predominantly studied in regard to ventricular electrophysiology [22,25,36]. In rabbits, only a few studies have evaluated atrial electrophysiology and showed a higher incidence of delayed afterdepolarizations, larger late sodium current, larger calcium transients and larger sarcoplasmic reticulum calcium contents in atrial cardiomyocytes isolated from male left atria [37]. Interestingly, no differences were observed in cardiomyocytes isolated from right atria [37]. In tissue preparations from male rabbits’ pulmonary veins or left atrium a higher spontaneous beating rate and incidence of burst firing as well as longer action potentials have been observed [38]; however, it has not been studied whether these alterations result in an increased AF susceptibility in male rabbits.

Despite numerous advantages of small animal species and their usefulness, especially in studying fundamental proarrhythmic pathways, the translatability into clinical application remains challenging, mainly because of substantial differences in size, anatomy, and electrophysiology, especially in mice [27,39]. Thus, findings from small animal models need to be confirmed in large animal models prior to clinical application in human patients [27,39]. In the presented study, we used a pig model to investigate sex-related differences in AF susceptibility, as pigs are a well-established species for translational cardiovascular research with great advantages over other commonly used large animal species, such as dogs and sheep [40,41]. Pigs share a vast number of physiological and anatomical similarities with humans. More importantly, humans and pigs express the same major ion channels in cardiac myocytes, resulting in similar action potential length and morphology, in both the atria and ventricles [27,39,42], making the pig an ideal model for the study of electrophysiology [27]. We investigated a porcine ischemic heart failure model, since one of the main triggers leading to AF is acute myocardial infarction (AMI) [26,43] with up to 25% of patients developing AF during or after AMI [43]. AMI commonly causes ischemic heart failure with reduced ejection fraction (HFrEF) [44], an effect that was seen in both sexes in our study. Our model therefore closely resembles the clinical situation of patients with AMI and allows an in-depth analysis of sex effects on ischemia-mediated atrial arrhythmogenicity.

In our study, IHF resulted in an enhanced susceptibility for AF in both sexes with male pigs showing significantly more frequent and a trend towards longer AF episodes compared to female animals. These findings are consistent with studies in mice demonstrating an enhanced AF susceptibility in male mice [31,32,45]. Furthermore, these data are in line with a clinical study including 27,512 patients (42% women and 58% men) that demonstrated a similar AF incidence in female and male patients during the 30-day monitoring period (50.2% vs. 49.8%) with a significantly increased AF episode duration and overall AF burden in men [46].

We revealed differences in several quantitative ECG traits between male and female pigs. Female control pigs showed a significantly lower resting heart rate and longer QRS intervals compared to male pigs, possibly due to a correlation between body weight/size with heart rate. It has been demonstrated in humans that a higher BMI positively correlates to a longer QRS interval [47]. Similarly, the higher QRS interval in the female control pigs might be attributed to their higher average body weight compared to the age-matched male control animals. In their book, *Swine in the Laboratory,* M. Swindle and A. Smith published ECG parameters of several minipig breeds such as the Göttingen or Hanford Minipig. Minipigs show higher heart rates, shorter QRS and QTc intervals compared to large domestic swine, but no significant differences between males and females were observed [48]. Overall, the higher resting heart rates and shorter QRS/QTc intervals correlated with a lower mature body weight, ranging from 12–45 kg in miniature breeds compared to large domestic swine breeds. In humans, women have a higher resting heart rate than men, which may be due to their smaller heart size and lower body weight [49]. Also, a longer QTc interval is seen in women [47], however, we did not observe this in our female pigs. Nevertheless, as we studied juvenile pigs, of which only some had already reached sexual maturity, our results are in line with studies in humans, which suggest that the QTc interval changes throughout development, probably due to different testosterone levels: in neonates and young children, no differences in QTc interval between sexes have been observed [19,50]. During puberty it is shortened in males, and in late adulthood (from 50 years) the QTc interval slowly prolongs and reaches a similar duration as in women [20], therefore we cannot rule out that the QTc interval may still change throughout the pigs’ lifetime.

Both in control and IHF pigs, the SNRT/BCL was longer in female pigs. IHF even caused a prolongation above 160%, which is considered pathologic in humans [51], indicating relevant sinus node dysfunction (SND) whereas in male IHF pigs, SNRT/BCL remained within a physiologic range. The SNRT/BCL is directly dependent on the heart rate, therefore it remains debatable whether the prolonged SNRT/BCL measured in females is true or false-positive due to the lower heart rate seen in female pigs. However, even the uncorrected SNRT is elevated in female control pigs (Appendix A), indicating a true finding in females. Separating animals according to those with and without AF reveals that in female animals with AF SNRT/BCL was slightly elevated compared to females without AF, both in the control and IHF group (Appendix A). Consistently, male pigs with and without AF had lower SNRT/BCL values than the female cohorts and differences between the male cohorts were not visible. However, the sample size per subgroup was low and did not allow reliable statistical analysis as to whether SND and AF correlate. In humans, SNRT is shorter in women but does not significantly differ from that in men when corrected to the intrinsic heart rate [52,53]. However, when affected, women more frequently suffer from SND such as sick sinus syndrome (SSS) compared to men [15,53]. SND and AF are closely linked as they often coexist (40–70% of patients diagnosed with SND already have AF), and SND markedly increases the risk to develop AF [54]. One study even suggests that a prolonged corrected SNRT (cSNRT) can be used as a predictor for paroxysmal AF recurrence following radiofrequency catheter ablation [55]. However, despite showing SND, female IHF pigs did not show more AF than male IHF pigs. To our knowledge, sex-related differences regarding sinus node function have not been investigated in any other animal model so far, therefore it remains unclear, whether a prolonged SNRT in females is physiologic in pigs. However current evidence demonstrates that pigs in general have a shorter SNRT than humans [52,56,57], supporting the interpretation of prolonged SNRT in our female IHF pigs. This suggests, female IHF pigs showing both SND (indicated by prolonged SNRT/BCL) and enhanced susceptibility for AF compared to controls closely resemble the situation in human patients and may establish a valuable model to investigate the mechanistic links between both diseases, which are still largely unknown so far.

As structural remodeling is one of the major hallmarks of AF pathophysiology, we investigated whether myocardial ischemia induced the development of interstitial fibrosis. We found significantly increased levels of LA fibrosis in both male and female IHF pigs as well as LV fibrosis in female IHF pigs indicating structural remodeling following myocardial injury. A recent study using late gadolinium-enhanced magnetic resonance imaging (LGE-MRI) for quantification of atrial fibrosis has shown increased atrial fibrosis in women with AF [58]. However, this may be an effect of age rather than sex since the women included in the AF group were significantly older than the male patients and since it has been demonstrated before that fibrosis gradually increases with advanced age [51,58]. In another study, Li et al. showed enhanced fibrosis in pulmonary vein sleeves from women but not men with long-standing persistent AF (LSP-AF) [59]. Gene and protein expression studies from these patients further indicated differences between men and women, specifically, an up-regulation of the TGFβ/Smad3 pathway that was observed in women [59]. However, in the studies conducted by Akoum et al. and Li et al., the average age of participants with AF was above 50 years, suggesting that age is an important contributor to the AF substrate [58,59], an effect that is probably not yet seen in our young, still growing pigs. Additionally, the left atrial tissue samples analyzed in our study do not depict the whole atria, as was shown in the above-mentioned LGE-MRI study. We therefore cannot rule out sex-related differences in regard to the distribution of fibrosis throughout the atrium including the pulmonary vein sleeves.

## 5. Limitations and Outlook

Although porcine models can generally serve as close-to-human models for cardiac research, there are, in contrast to small animal models such as mice, only a few genetically modified swine models available, as genetic engineering has proven to be more difficult in large animals [60]. This limits the spectrum of human diseases for which the pig as an experimental species can be used to dietary-, medically- or instrumentally-induced disease models. However, with ongoing advancement of genetic engineering and CRISPR-Cas, more and more genetically modified swine models have been developed recently [27,60,61].

In humans, substantial differences in electrophysiology and arrhythmogenesis between men and women can be observed. In our study, however, we observed only moderate differences between male and female pigs, which can potentially be attributed to the age of our pigs. We included young, still growing pigs whereas sex differences in human patients are mostly shown for adults or even in an aged population. As the influence of the biological sex on cardiac electrophysiology is probably mainly driven by sex hormones with varying blood levels over time, it remains unclear whether the differences found in our pigs can be explained by fluctuations of hormone levels, since there are only limited data available on sex hormone levels or receptor distribution in pigs over time. Therefore, studies on older pigs should be performed in the future as well. This can be challenging in our selected swine breed, as mature domestic wild type swine can weigh up to 350 kg and would require specialized animal facilities and well-trained staff. An alternative could be to use minipig breeds, such as the Göttingen Minipig, for long-term studies.

Reports on physiological ranges of many parameters in pigs are inconsistent, as animals of various breeds and ages are used, and often do not differ between males and females due to low sample sizes. Although our overall sample size is small on a clinical scale, the group sizes in our study exceed most study sizes in large animal research, thus showing robust results. Yet, to fully exploit the potential of our pig model, more extensive study protocols could verify the differences between the male and female ECG parameters and EP properties, such as sinus node function.

## 6. Conclusions

Our study demonstrates that ischemic heart failure increases the vulnerability for atrial fibrillation (AF) in pigs and thus confirms this model as a clinically highly relevant close-to-human model for atrial arrhythmogenesis. In this model, we observed biological differences between male and female swine, most importantly a significantly prolonged sinus node recovery time in female IHF pigs, mirroring the situation in female patients who suffer more frequently from sinus node disease. In sum, this model seems to be ideally suited to further investigate IHF-related atrial arrhythmogenicity in a close-to-human environment, even mimicking sex-related effects on electrophysiology.

## Figures and Tables

**Figure 1 cells-12-00973-f001:**
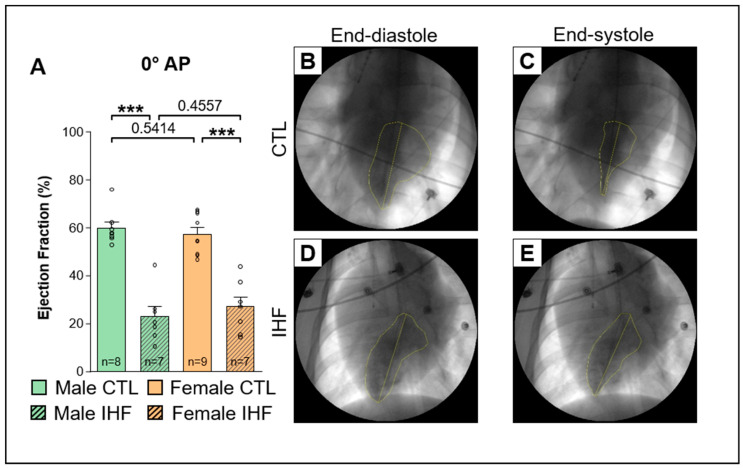
Ejection fraction (EF) assessed by laevocardiography at 130 bpm at 0° AP (anterior posterior). (**A**), Ejection Fraction. (**B**,**C**), Representative images of a control pig at end-diastole (**B**) and end-systole (**C**). (**D**,**E**) Representative images of an IHF pig at end-diastole (**D**) and end-systole (**E**). *CTL*, control animals without ischemic heart failure; *IHF*, animals with ischemic heart failure. Bar graphs represent Mean + SEM, grey circles represent data of individual pigs. Mann-Whitney-U Test. *** *p* < 0.001.

**Figure 2 cells-12-00973-f002:**
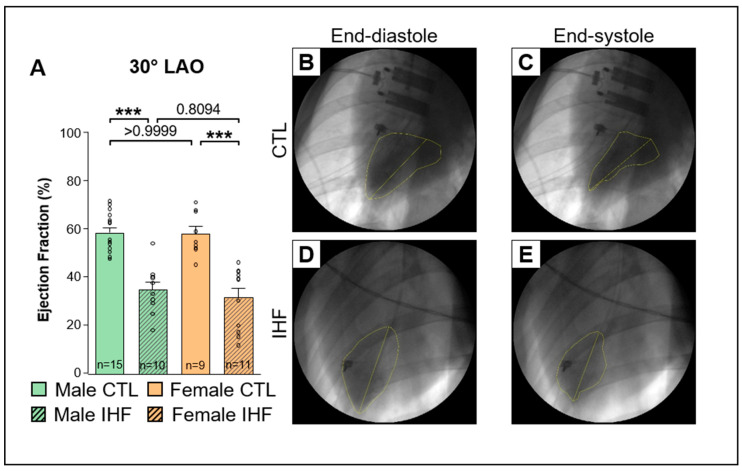
Ejection Fraction (EF) assessed by laevocardiography at 130 bpm at 30° LAO (left anterior oblique). (**A**), Ejection Fraction. (**B**,**C**) Representative images of a control pig at end-diastole (**B**) and end-systole (**C**). (**D**,**E**) Representative images of an IHF pig at end-diastole (**D**) and end-systole (**E**). *CTL*, control animals without ischemic heart failure; *IHF*, animals with ischemic heart failure. Bar graphs represent Mean + SEM, grey circles represent data of individual pigs. Mann-Whitney-U Test. *** *p* < 0.001.

**Figure 3 cells-12-00973-f003:**
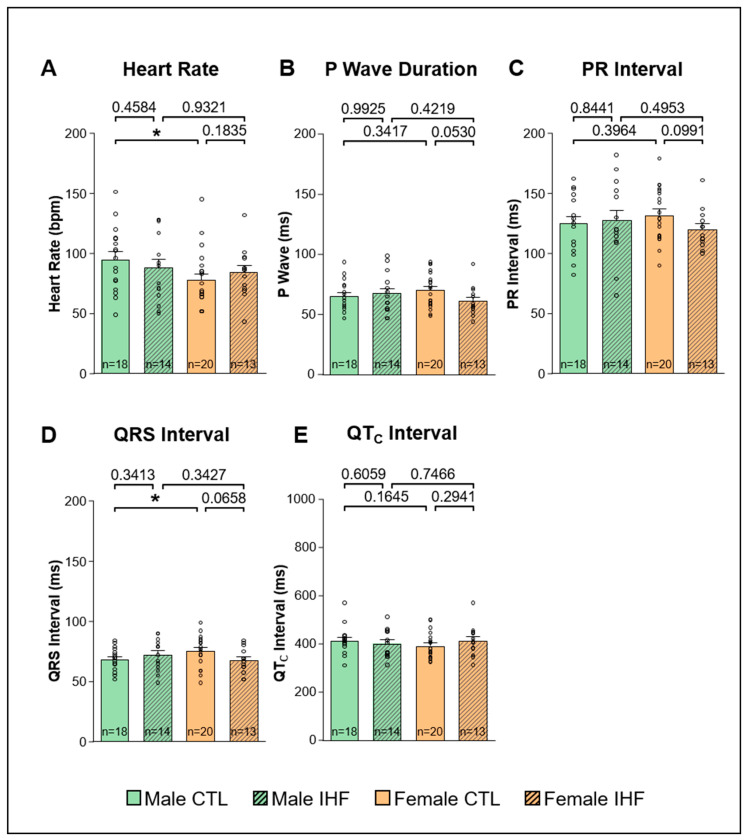
ECG parameters. (**A**), Heart Rate. (**B**), P Wave Duration. (**C**), PR Interval. (**D**), QRS Interval. (**E**), Corrected QT Interval (calculated according to Bazett). *CTL*, control animals without ischemic heart failure; *IHF*, animals with ischemic heart failure. Bar graphs represent Mean + SEM, grey circles represent data of individual pigs. Mann-Whitney-U Test. * *p* < 0.05.

**Figure 4 cells-12-00973-f004:**
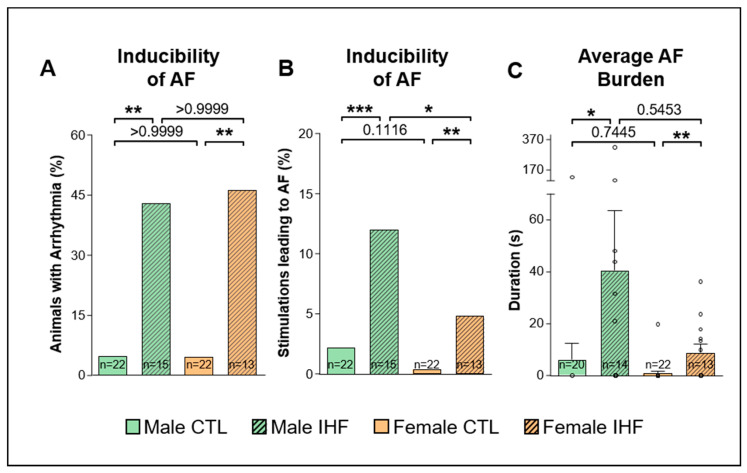
Inducibility of Atrial Fibrillation (AF) (**A**), Percentage of pigs with AF episodes longer than 10 s. (**B**), Percentage of burst stimulations leading to AF episodes longer than l0 s. (**C**), Average AF burden. *CTL*, control animals without ischemic heart failure; *IHF*, animals with ischemic heart failure. Bar graphs represent Mean + SEM, grey circles represent data of individual pigs. Fisher’s Exact Test (**A**,**B**) and Mann-Whitney-U Test (**C**). * *p* < 0.05; ** *p* < 0.01; *** *p* < 0.001.

**Figure 5 cells-12-00973-f005:**
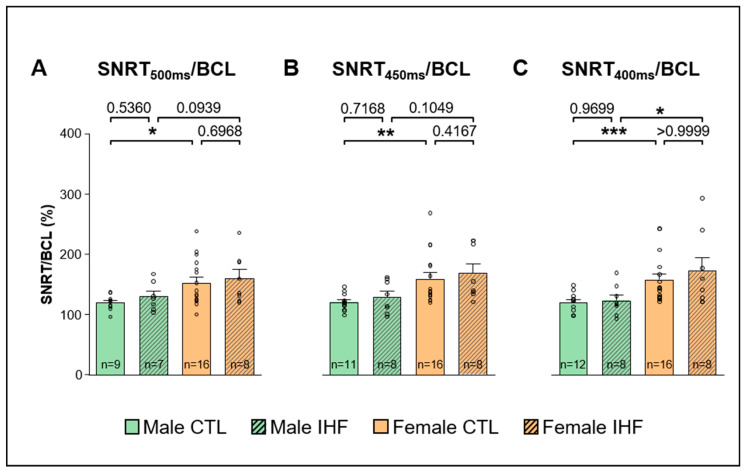
Corrected Sinus Node Recovery Time (SNRT/BCL). (**A**), SNRT/BCL at a pacing cycle length of 500 ms. (**B**), SNRT/BCL at a pacing cycle length of 450 ms. (**C**), SNRT/BCL at a pacing cycle length of 400 ms. *CTL*, control animals without ischemic heart failure; *IHF*, animals with ischemic heart failure. Bar graphs represent Mean + SEM, grey circles represent data of individual pigs. Mann-Whitney-U Test. * *p* < 0.05; ** *p* < 0.01; *** *p* < 0.001.

**Figure 6 cells-12-00973-f006:**
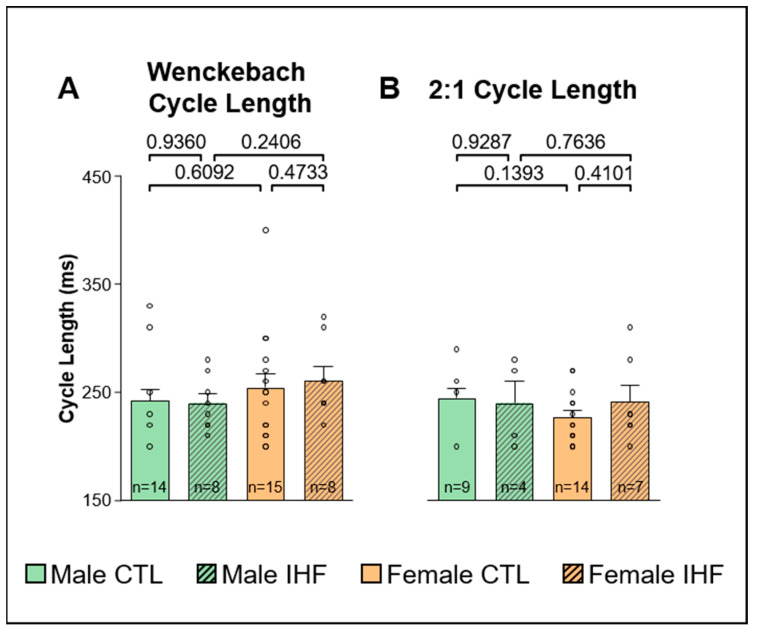
Atrioventricular conduction properties. (**A**), Wenckebach cycle length. (**B**), 2:1 atrioventricular conduction cycle length. *CTL*, control animals without ischemic heart failure; *IHF*, animals with ischemic heart failure. Bar graphs represent Mean + SEM, grey circles represent data of individual pigs. Mann-Whitney-U Test.

**Figure 7 cells-12-00973-f007:**
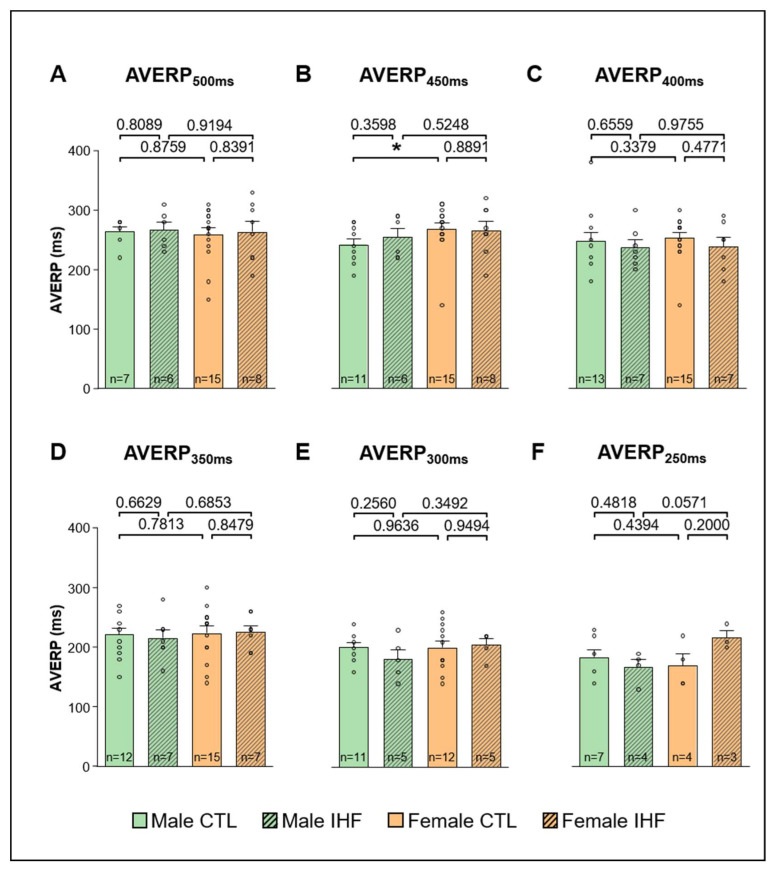
Atrioventricular Effective Refractory Period (AVERP). (**A**), AVERP at a basic cycle length of 500 ms. (**B**), AVERP at a basic cycle length of 450 ms. (**C**), AVERP at a basic cycle length of 400 ms. (**D**), AVERP at a basic cycle length of 350 ms. (**E**), AVERP at a basic cycle length of 300 ms. (**F**), AVERP at a basic cycle length of 250 ms. *CTL*, control animals without ischemic heart failure; *IHF*, animals with ischemic heart failure. Bar graphs represent Mean + SEM, grey circles represent data of individual pigs. Mann-Whitney-U Test. * *p* < 0.05.

**Figure 8 cells-12-00973-f008:**
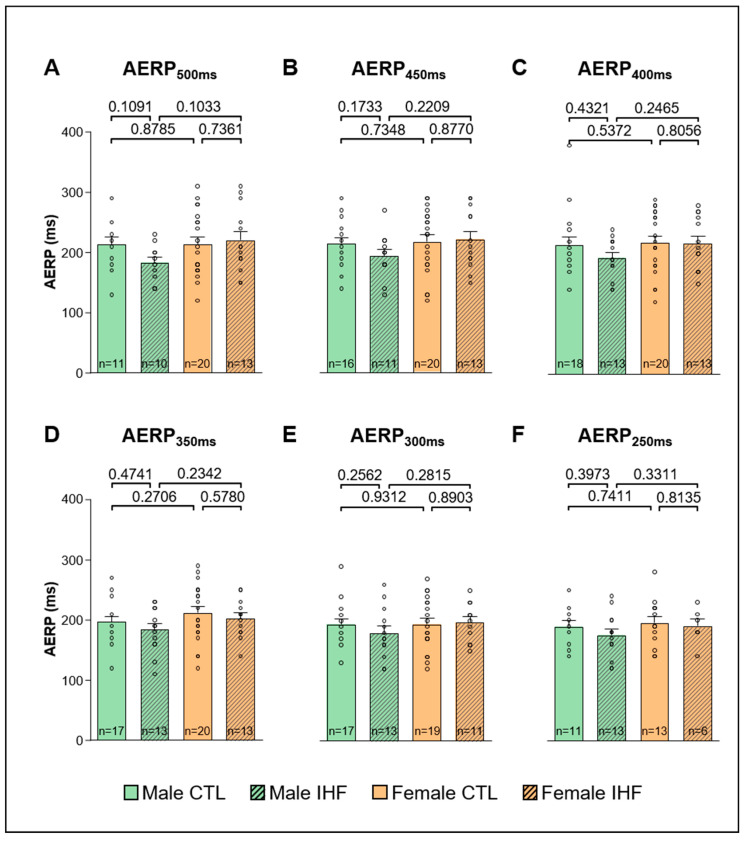
Atrial Effective Refractory Period (AERP). (**A**), AERP at a basic cycle length of 500 ms. (**B**), AERP at a basic cycle length of 450 ms. (**C**), AERP at a basic cycle length of 400 ms. (**D**), AERP at a basic cycle length of 350 ms. (**E**), AERP at a basic cycle length of 300 ms. (**F**), AERP at a basic cycle length of 250 ms. *CTL*, control animals without ischemic heart failure; *IHF*, animals with ischemic heart failure. Bar graphs represent Mean + SEM, grey circles represent data of individual pigs. Mann-Whitney-U Test.

**Figure 9 cells-12-00973-f009:**
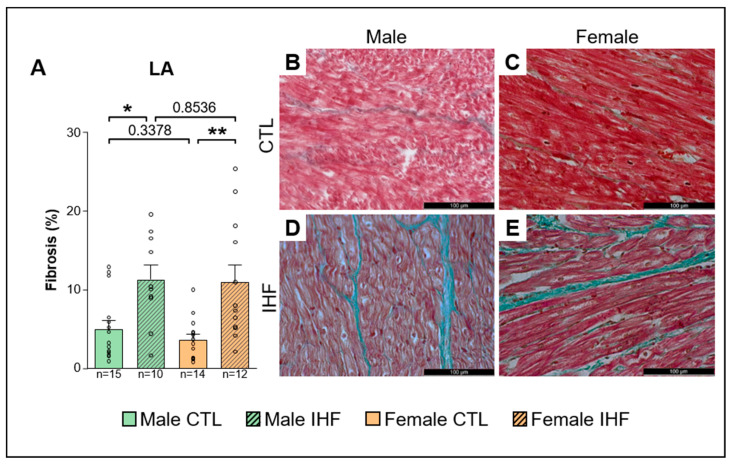
Interstitial fibrosis assessed by Masson‘s trichrome staining in the Left Atrium (LA). (**A**), LA Fibrosis. (**B**,**C**), Representative images of control pigs in the LA of males (**B**) and females (**C**). (**D**,**E**), Representative images of IHF pigs in the LA of males (**D**) and females (**E**). *CTL*, control animals without ischemic heart failure; *IHF*, animals with ischemic heart failure. Bar graphs represent Mean + SEM, grey circles represent data of individual pigs. Mann-Whitney-U Test. * *p* < 0.05; ** *p* < 0.01.

**Figure 10 cells-12-00973-f010:**
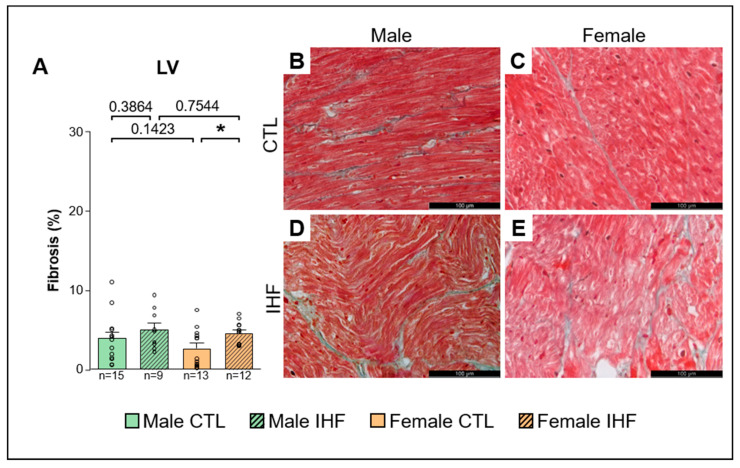
Interstitial fibrosis assessed by Masson‘s trichrome staining in the Left Ventricle (LV). (**A**), LV Fibrosis. (**B**,**C**), Representative images of control pigs in the LV of males (**B**) and females (**C**). (**D**,**E**), Representative images of IHF pigs in the LV of males (**D**) and females (**E**). *CTL*, control animals without ischemic heart failure; *IHF*, animals with ischemic heart failure. Bar graphs represent Mean + SEM, grey circles represent data of individual pigs. Mann-Whitney-U Test. * *p* < 0.05.

## Data Availability

Data is available upon reasonable request to the corresponding author.

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
