# Peer review of "Effects of Sex on the Susceptibility for Atrial Fibrillation in Pigs with Ischemic Heart Failure"

_cells, 2023, doi:10.3390/cells12070973_

Round 1
Reviewer 1 Report
Authors addresses a very relevant topic: sex effect on atrial electrophysiology. There is a delay in age-adjusted incidence of AF in women. Much is known about sex-related differences in electrophysiology in the ventricles. However situation in the atrium is almost completely unknown. Therefore, the study is very welcomed. Many of the authors are well-established experts in the field of cardiac electrophysiology, some of them even in the field of animal models of AF. Thus the study is well planned, number of experiments is high enough to allow meaningful interpretation. Results are convincing and nicely demonstrated. The ms is well written. I see some room for improvement regarding presentation and discussion of the data.
Major points
1. Key message
What is the most important finding of that study? In principle effects of sex can be mediated by a genomic or non-genomic mechanism. If I understood the authors correctly the pigs in that study were prepubertal? This would argue that there are not too much genomic effects between male and females in atrial electrophysiology (for controls [no IHF] even not a difference in AF inducibility/burden?
Minor points
1. Methods (l. 99)
Atropine wasn´t used for sedation? Please check.
2. Flow of results
I would propose to give first data on ECG and focus afterwards on data on invasive atrial electrophysiology.
3. AF inducibility (Figure 3)
Figure 3 is nice bit could give rise to some confusion. I would suggest to give data shown in Figure 3A as a sentence in the text to avoid unnecessary complication.
4. Sinus node recovery time (l. 221
Please indicate how % values were calculated.
5. Advantage of pigs as a model (l. 344)
I clearly see the enthusiasm of the authors for the animal model they used. However, in order to avoid confusion of potential readers authors should discuss limitations of pig as an electrophysiological model (correct me if I am wrong).
6. N numbers
Are all experiments performed with the same n numbers. Maybe it would be appropriate to give nu member in the legends or figures?
7. Recent study in human tissue
I found a recent study worth to mention: DOI: 10.1016/j.yjmcc.2023.01.006. Please check.
Author Response
Please find our point-by-point response enclosed.

Reviewer 2 Report
This paper proposed by Valerie Pauly et al, entitled " Effects of Sex on the Susceptibility for Atrial Fibrillation in Pigs with Ischemic Heart Failure" they investigated the potential impact of sex on the susceptibility for atrial fibrillation (AF) using a pig model of ischemic heart failure (IHF) and observed a significantly increased vulnerability for AF in IHF pigs of both sexes. This paper is very interesting and novel.
1) Why did not the authors Show/present the results of the hemodynamic variables between both the sexes in IHF pigs?
Author Response

(The authors gave the same response as above.)

Reviewer 3 Report
I would like to congratulate the authors on the selection of this novel and the interesting aspect of cardiology. The authors have provided good evidence to support their conclusion with well constructed and meticulously written manuscript.
However, I would like to bring attention to the following points
1. Please elaborate statistical section on the methodology
2. In the results section provided results in graphical format can be helpful for readers
3. In the limitations section more succinct presentation is needed.
Author Response

(The authors gave the same response as above.)
